# The Effect of Risk Communication on Public Behavior to Non-Conventional Terrorism—Randomized Control Trial

**DOI:** 10.3390/ijerph19010342

**Published:** 2021-12-29

**Authors:** Moran Bodas, Morel Ragoler, Yossi Rabby, Esther Krasner

**Affiliations:** 1The Gertner Institute for Epidemiology and Health Policy Research, Sheba Medical Center, Tel Hashomer, Ramat-Gan 5262100, Israel; morelr@gertner.health.gov.il; 2The Department of Emergency & Disaster Management, School of Public Health, Sackler Faculty of Medicine, Tel-Aviv University, Tel-Aviv-Yafo 6997801, Israel; 3CBRN Defense Division, Ministry of Defense, HaKirya, Tel-Aviv-Yafo 6473424, Israel; jos.rabby@gmail.com (Y.R.); esther3@bezeqint.net (E.K.)

**Keywords:** non-conventional terrorism (NCT), behavior, risk communication, fake news, randomized control trial (RCT)

## Abstract

Non-conventional terrorism (NCT) incorporates an extended dimension of uncertainty that can lead to fear among the public. Health officials have an unsubstantiated assumption that thousands will seek treatment in hospitals following NCT. This study aims to examine public behavioral intentions in the case of NCT and the effect of risk communication on intents. An online randomized controlled trial was conducted among 1802 adult participants in Israel. Threat perception and behavioral intent before and after exposure to hypothetical NCT scenarios were assessed stratified to the type of media, exposure to rumors, and risk communication. The majority (~64%) of participants are aware of the NCT threat. Almost half (45%) of participants indicated a “high” or “very high” chance of seeking medical attention following an NCT incident. Regression analysis suggests that the odds of participants exposed to risk communication to report an elevated intent of seeking medical attention were 0.470 (95% CI: 0.359, 0.615) times that of participants not exposed to risk communication, χ^2^ = 30.366, *p* < 0.001. The findings demonstrate the importance of effective risk communication in reducing undesired public behavior during NCT crises. Efforts must be invested to create a robust risk communication infrastructure to allow the proper management of possible NCT incidents.

## 1. Introduction

Non-conventional terrorism (NCT), also known as chemical, biological, radiological, or nuclear (CBRN) terrorism, entails the use of such substances to gain political goals and poses a serious challenge to crisis managers from a public health point of view. Being a form of intergroup threat, NCT is inherently infused with elevated anxiety [1]. This attribute is described in numerous relevant social psychology frameworks, as early as the realistic group conflict theory [2], and, more recently, Stephan and Stephan’s integrated group threat theory [3].

Scenarios involving CBRN are infused with an extended dimension of uncertainty drawn from the lack of familiarity with such incidents. Consequently, the use of CBRN agents in a terrorism context is likely to cause emotional distress and amplify fear and anxiety responses by the public [4,5,6,7,8].

Several factors have been described as being associated with an elevated perception of threat by the public. In particular, NCT incidents may have serious implications to public health due to several inherent characteristics, including (a) threat perceived as external, e.g., terrorism, (b) being an “out-of-the-blue” occurrence, (c) being manmade, (d) an unfamiliar threat, (e) their ability to induce severe and unusual medical conditions, and to (f) place children at risk [4].

NCT incidents can cause the public to experience emotional, psychological, social, and behavioral responses far beyond other threats, which last longer and have more serious implications in daily life [5,6,7,8,9]. Even small scale NCT incidents are capable of generating confusion, fear, stress, and anxiety that can harm the well-being of the public [4,10,11,12,13,14].

Many factors are associated with threat perception and how people react to threats, especially those perceived as existential [15]. Perception of likelihood, severity, threat intrusiveness (i.e., the extent to which a person associates himself and his own risk with a given threat), control, fear, optimism, and prior experience are some examples of such factors [16,17]. Of similar importance are components of coping mechanisms with the threat, such as the perceived efficacy of life-saving instructions, their purpose, self-efficacy to comply with them, and their cost-benefit [18].

According to the stress and coping theory by Lazarus and Folkman [19], upon a real and immediate danger to life, people respond with a sharpened assessment of personal threat by applying threat appraisal. The first of these is “risk as feeling”, which relies mostly on an intuitive, immediate, rapid emotional response to the threat. The second is “risk as analysis”, which employs a more logical and reasoned evaluation of the risk. Naturally, people are more prone to use the first rather than the latter in the absence of appropriate risk communication [10,16,20,21].

Research has demonstrated the importance of effective and rapid risk communication on minimizing negative emotional responses by the public in non-conventional incidents [7,14,16,20,22]. Risk communication research comprises elements of social, cognitive, and economic psychology, drawing on areas such as health promotion, media communications, and disaster management and applying principles from each [23]. Subsequently, many theories and approaches to risk communication exist. Some prominent among these include the learning theory [24], the social cognitive model [25], the communication-persuasion matrix [26], and Covello’s organizing models, including the risk perception model [27] and the extended parallel process model [28], and Mileti and Peek’s social psychology of public response to warnings [29].

In the context of NCT, risk communication is used to improve public understanding of the threat [30], to keep the public informed, to manage fear [27], and to encourage cooperation with, and adherence to instructions given by authorities to allow the public to protect themselves in the event [31]. To be effective, risk messaging requires more than just knowledge of the risk, but should also allow the recipient to feel empowered to act, be honest and open, and should adopt cultural and demographic requirements, including language and communication style [32].

Of note, studies have shown that communicating the true dimensions of the threat, the actions being taken by authorities and the immediate risk to the population have a profound influence on protective behavior among the public [7,27]. For example, following an effective and coordinated distribution of risk communication in the aftermath of the 2006 poisoning of the former Russian spy, Litvinenko, in London, which posed a public health concern not only in the UK but among the guests of the hotel where the spy was poisoned worldwide, Londoners were found to perceive the threat to their personal health as low, at 11% [11]. This was achieved through proper media management, which communicated that the situation was under control and that the risk directly posed to the public was limited. Consequently, the extent to which the public health authorities had to support members of the public concerned about potential exposure was very low and allowed the public health system to manage the situation effectively [10].

Failure to provide effective risk communication during a CBRN incident can lead to undesired public behavior, which can place more people at risk [10,16]. For instance, following the Tokyo Sarin attack in 1995, people continued to seek medical attention for days following the incident. According to Beaton et al. [33] (p. 108), “timely and accurate risk communication might have reduced the number of worried well seeking medical treatment in the aftermath of a mass casualty event such as those seeking treatment in Tokyo two days or more following the attack”. Another example is the 2001 Anthrax (AMERITHRAX) incident in the USA, where 5% of Americans acquired antibiotics against public health instructions, and 20% took the drug as preemptive medicine without a proper medical prescription [7,16,34].

Making things worse, the vacuum caused by a lack of effective risk communication is often filled by rumors and “fake news”, often distributed through social media outlets [35,36]. This is especially true in situations of increased uncertainty, such as that existing in NCT incidents. Under such circumstances, it is desired that the public take action where it is recommended for protection. Sadly, rumors and misinformation may lead to injury and even death, as proposed by a recent paper that examined rumors, stigma, and conspiracy theories during the Novel Corona Virus (COVID-19) pandemic [37]. According to Islam et al. [37], approximately 800 people lost their lives, 5876 have been hospitalized and 60 lost their eyesight after drinking methanol, believing it to be a cure for the disease.

In the absence of effective government-directed risk communication, the public can become distrusting, which could further exacerbate undesired and risky behavior [7,25,38,39,40]. For example, during the Swine Influenza pandemic of 2009, public discourse was swamped with rumors about vaccine safety issues. As a result, refusal rates of vaccination skyrocketed and demand for costly vaccines was very low [38,41].

Previous studies have provided guidelines for effective risk communication during CBRN crises. These include having a simple and unified message that addresses actual proportions of the risk and what the public should expect in the near future [7,10,20,42]. In addition, establishing public trust is of utmost importance for risk communication to take precedence over rumors and “fake news”. Communicating authorities can foster trust in their ability to provide accurate information and effective actions through their messages by maintaining openness and honesty. This includes stating that “we do not know at this point of time, but are working relentlessly to collect more information” [4,7,10,20,40,42,43].

Israel is faced with a multitude of terroristic threats, including CBRN terrorism. In planning for such incidents, the working assumption of the Israeli Ministry of Health (and other health officials worldwide) is that the public will flock to hospitals in pursuit of medical attention following an NCT incident. This assumption is unsubstantiated [44]. Nonetheless, there are pieces of evidence in the literature to support the notion that following NCT incidents healthcare systems may become overcrowded by many lightly wounded or anxious patients (formerly known as “worried well”) [45,46]. For example, according to Stone [47] (p. 1): “these worried well patients may comprise as many as 20 times the number of legitimate patients and may become one of the most difficult aspects of dealing with (such) events”.

The purpose of this study was to explore public behavioral intentions in the case of NCT, particularly the intention to seek medical assistance in hospitals following an NCT scenario. This study examined the effect of risk communication on such intent, as well as the effect of the type of media and the presence of rumors/fake news on behavioral intent. The novelty of this study is two-fold: (a) to the best of our knowledge, it is the first attempt to assess public intention to overcrowd hospitals in case of an NCT, and (b) it explores this intention based on a matrix of type of media exposure, exposure to fake news, and the possible effect of timely risk communication on preventing unwanted behavior. We hypothesized that (H1) participants exposed to the scenarios on social media will demonstrate an elevated perception of the threat compared with those exposed to classic media reports. It is assumed that exposure to rumors and fake news in the social media context will increase the perceived threat compared to study groups in which no rumors were used. In addition, we hypothesized (H2) that participants exposed to risk communication and instructions will demonstrate less intent to undertake undesired behavior (namely seek unwarranted medical care) compared to the control group. It is assumed that exposure to risk communication can negate the negative psychological effects that can lead to undesired behavior.

## 2. Materials and Methods

### 2.1. CONSORT Statement

This manuscript was prepared in accordance with the CONSORT guidelines for randomized controlled trials [47].

### 2.2. Study Type

This study was a randomized control trial based on online recruitment and randomization of participants into study control and intervention groups. The study was carried out over three days in May 2020. The choice to perform this study online was in light of the multitude of visual tools used in the study, including media reports, risk communication instructions, and rumors/fake news.

### 2.3. Population and Sample

The study population was the adult (over 18 years of age) population of Israel. The minimum sample size for this population is estimated at ~9,000,000 people with 95% confidence and 5% marginal error is 385 people [48]. This study included a large number of comparison groups. Thus, to ensure a minimal sample size of at least 100 participants in each study group the final sample size was 1802 participants. The sample was representative of the target population based on predetermined quotas (Table 1).

Participants were recruited from the internet panel of iPanel in a manner that ensured their anonymity. Since 2006, the iPanel provides an online platform for a wide variety of information collection services, including polls and public opinion surveys. It adheres to the stringent standards of the European association for market, social, and opinion researchers (ESOMAR). Panelists of the iPanel are compensated for their participation in surveys and polls, including the current study.

### 2.4. Randomization and Study Design

Participants were randomly allocated from a pool of panelists according to predetermined socio-demographic quotas and assigned to three branches of the study representing three different CBRN scenarios (chemical, biological, and radio-nuclear). Participants who consented to the informed consent statement only were included in the study. The informed consent statement informed participants that they would be interacting with hypothetical yet plausible NCT scenarios. Next, for each scenario, participants were further randomized into two study branches representing two different media types (classic online news outlets and social media).

Within the classic media study branch, participants were randomly assigned to either a control group, in which no additional intervention following exposure to the CBRN scenario took place, and an intervention group in which risk communication and instructions were provided (see Appendix A for details). Within the social media study branch, participants were randomly allocated to three groups: control, exposure to rumors/fake news, and exposure to rumors/fake news plus risk communication (Appendix A). The rumor/fake news post on social media was designed to look as if a genuine person published it. The post exaggerated the threat and called people to distrust the authorities and seek medical attention as soon as possible. Figure 1 provides a flow chart of participants’ randomization into study groups.

To preserve uniformity across study branches, a similar scenario was used for all types of CBRN scenarios involving the spread of chlorine gas, white Anthrax powder, or a “dirty bomb” laced with radioactive material (for the chemical, biological, and radiological scenarios, respectively), onboard the train en-route from Nahariya in the north to Beer-Sheva in the south. The imaginary attack took place while the train was at a stop at the Tel-Aviv Central Station at around 16:00 (peak time). Screenshots of the news reports and rumor posts are available in Appendix B.

### 2.5. Tools and Variables

The primary tool used in this study was a questionnaire administered online. The questionnaire was developed and validated through consultations with an experts’ panel (data not published). The questionnaire was available to participants in two languages: Hebrew or Arabic, at their choice. Participants completed the questionnaire at two time points—before the intervention and immediately after.

#### 2.5.1. Primary Outcome

The primary outcome of this study was behavioral intent. Primarily, the study assessed intent to seek medical attention in hospitals in response to the NCT scenario on a 5-point Likert scale ranging from “very low” (assigned the score 1) to “very high” (5). In addition, participants were presented with a list of nine other behaviors (see the complete list in the results) to choose from. This was administered under two separate conditions: the first based on the NCT scenario as presented, and the second if, hypothetically, the scenario happened in the hometown of the participant. An overall “behavioral intent” score was calculated per participant. Participants were also allowed to add free text to indicate other behaviors they would intend to undertake in response to the scenario presented.

#### 2.5.2. Secondary Outcomes

The secondary outcomes assessed in this study included: (1) prior knowledge (1 item)—assessing previous acquaintance with the concept of NCT (“before you participated in this study, were you aware of the term “non-conventional terrorism?”); (2) threat perception (four items)—assessing the perception of NCT likelihood (“in your opinion, what are the chances of an NCT incident happening in Israel in the next year?”), threat intrusiveness (“assuming such incident happens, what are the chances of you or your family members being personally affected by it, e.g., injury or death?”), and severity (“assuming such incident happens, how severe do you think it will be to the routine of life of the entire society and your family’s routine?”—two items). These items were adopted from previous studies dealing with the similar context of armed conflicts in Israel [17,49,50]; (3) indifference to emergencies (one item)—assessing the extent to which a participant is concerned or indifferent to the concept of emergencies, as a possible confounder (“in general, to what extent are you influenced by emergencies in Israel?”); (4) perception of response (five items)—assessing situational concern from NCT (“to what extent are you worried about NCTs?”), trust in authorities (“to what extent do you trust the instructions of the emergency organizations during an emergency?”), and sense of preparedness (three items)—assessing knowledge of protective behavior, mental preparedness, and actual preparedness, based on Bodas et al. [17,49]; (5) interest in additional information—assessing participant’s wish to receive more information about NCTs. The majority of items were assessed using a 5-point Likert scale, except for the severity scale, which ranged from 1 (not at all), through 2 (slightly severe) and 3 (quite severe) to 4 (very severe).

Socio-demographic variables collected included gender, age, religion (Jewish versus non-Jewish), affiliation to religion (secular, traditional, religious, ultra-orthodox), place of residence, marital status, number of family members and children, level of education, level of income, and profession. All participants in this study were completely anonymous to the researchers.

### 2.6. Statistical Analysis

Statistical analysis was conducted using SPSS (ver. 25) and included both descriptive and inferred statistical analyses. Before performing statistical analysis, indices were generated and their validity was ensured with the Cronbach-alpha test. Statistical tests were chosen in accordance with the variable type and distribution. Differences in the proportion of categorical variables were assessed using the chi-square test. Differences between categorical and continuous variables were conducted using Student’s *t*-tests for dependent and independent samples, ANOVA test, or non-parametric equivalences (Mann–Whitney or Wilcoxon tests), according to variable distribution or sample size. Association between two or more continuous variables was done using Spearman correlation, with Bonferroni correction for multiple comparisons. In all statistical analyses, a *p*-value of 0.05 or smaller was deemed as statistically significant, except where correction for multiple comparisons was done.

In order to predict the primary outcomes (seeking medical attention at the hospital), an ordinal regression model was used. The analysis only included variables that were found to be associated with the dependent variable in univariate analysis, following negation of multi-collinearity. The model included the following variables: gender, age, indifference to emergencies, perception of likelihood, perception of severity (two items), threat intrusiveness, concern from NCT, trust, sense of preparedness, exposure to rumors (yes/no), exposure to risk communication (yes/no), interest in additional information, and the total number of planned behaviors.

## 3. Results

### 3.1. Prior Knowledge of NCT and Interest in More Information

Of the study sample, 1131 (62.8%) reported having prior knowledge of the concept of non-conventional terrorism (NCT). Prior knowledge ranged from 59% in the biological scenario to 65% in the radiological scenario with no statistical significance (χ^2^ = 5.38, df = 4, *p* = 0.250).

Men report more than women to have prior knowledge about NCT (70.5% compared to 55.4%, respectively), according to the chi-square test (χ^2^ = 38.38, df = 1, *p* < 0.001). Jews report more than non-Jewish (68.3% vs. 40.6%, respectively; χ^2^ = 78.89, df = 1, *p* < 0.001). Seculars are the largest group with prior knowledge (69.7%), followed by ultra-orthodox (63.7%), traditional (55.5%), and religious (53.1%) (χ^2^ = 45.77, df = 3, *p* < 0.001). No significant difference was observed between participants with an academic background (65.8%) and a non-academic background (60.8%) (χ^2^ = 1.28, df = 1, *p* = 0.260). The mean age of participants with prior knowledge was 41.47 (±14.31) years, which was significantly higher than the mean age of participants without prior knowledge (35.18 ± 12.99), according to independent samples *t*-test (t = 9.04, df = 1209.33, *p* < 0.001).

Of the overall sample, 1160 (64.4%) of participants expressed their interest to receive more information about NCT preparedness following their participation in the research. The greatest interest was among participants of the radiological scenario (66.7%), followed by the chemical (64.1%), and the biological (62.4%). Non-Jewish participants expressed a higher interest than Jewish participants (69.1% vs. 63.2%; χ^2^ = 4.341, df = 1, *p* = 0.037). No statistical differences were observed for other socio-demographic variables.

### 3.2. Threat Perception of NCT

The majority of participants (40.6%) reported that emergencies in Israel have a mediocre personal effect on them. About 39% replied “high” or “very high”, and 21% reported “little” or “very little.” Women, non-Jewish, and seculars reported experiencing higher effects of emergencies compared to men, Jews, and non-secular participants (*p* < 0.001, *p* = 0.024, and *p* < 0.001 respectively).

Participants were asked to report their perceptions of NCT likelihood, severity, intrusiveness, concern, trust in authorities, and sense of preparedness. Table 2 summarizes the results of this section. Interestingly, before the intervention, all three NCT scenarios were ranked similarly in terms of likelihood, severity, and threat intrusiveness. After the intervention, the chemical scenario was rated as likeliest to occur and the radiological as the most severe; however, levels of concerns from all three NCT scenarios were similarly high. In addition, no differences were found between participants allocated to classic and social media news reports either before the intervention or post-intervention (data not shown).

Trust in authorities declined following the intervention. In the overall sample (*N* = 1802), this measure dropped from a mean of 3.49 ± 0.89 before the intervention to 3.40 ± 0.41 following the intervention. The drop was statistically significant in all NCT scenarios, even when analyzed separately to participants who were exposed to risk communication (*p* = 0.018) and those who were not exposed to risk communication (*p* < 0.001). Trust was found to be associated with affiliation to religion, with non-seculars demonstrating higher trust levels than seculars (*p* = 0.003), but only among those who were not exposed to risk communication. Additionally, trust was also associated with education, with non-academics reporting higher trust levels than academics (*p* = 0.001), but only among those who were exposed to risk communication.

### 3.3. Effects of Exposure to Rumors and Risk Communication on Threat Perception

The data demonstrates no influence of exposure to rumors (“fake news”) on threat perception components in the biological and radiological scenarios. In the chemical scenario, rumors influenced the perception of NCT likelihood and severity to the participant’s family routine. While the perception of NCT likelihood decreased with exposure to rumors according to the ANOVA test (F = 5.05, *p* = 0.007), the perception of severity to the family increased with exposure to rumors (F = 5.09, *p* = 0.007). In both cases, Bonferroni’s correction revealed that the significance is attributed to the difference between the group exposed to the rumor and the control.

Among participants who were allocated to the classic media report, no significant effect was observed for exposure to risk communication on threat perception components. The only exception to this was the perception of NCT severity to the family’s routine in the radiological scenario. In this case, the mean change in the perception of severity following the intervention was −0.18 (±0.81SE) in the group exposed to risk communication compared to a mean change of +0.02 (±0.61SE) in the control group (t = −2.164, df = 220.83, *p* = 0.032).

Grouping together all participants from all scenarios based on their exposure to risk communication, the data show no statistical significance in threat perception components before the intervention between exposed and non-exposed participants (data not shown). However, following the intervention, those exposed to risk communication (*N* = 721) demonstrated the reduced perception of severity than those who were not exposed to risk communication, in both perceptions of severity to family (3.01 ± 0.80 vs. 3.09 ± 0.76, respectively; t = 2.139, df = 1800, *p* = 0.033) and severity to Israeli society (2.77 ± 0.82 vs. 2.85 ± 0.79, respectively; t = 2.237, df = 1800, *p* = 0.025).

### 3.4. Behavioral Intent

Participants were asked to indicate the likelihood that they would seek medical attention in a hospital if the scenario presented to them happened in reality. Table 3 summarizes the results of this section. In all scenarios, ~15% of participants indicated the highest option of “very high”. An additional ~30% indicated the second-highest option “high”.

Women report greater intent to seek medical attention than men (3.30 ± 1.21 vs. 3.16 ± 1.20; t = 2.44, df = 1800, *p* = 0.015). No significant differences were observed across religion, affiliation to religion, education, or age (data not shown).

Seeking medical attention in hospitals is positively associated with numerous threat perception components, including the perception of likelihood (r = 0.180), threat intrusiveness (r = 0.252), severity to society (r = 0.167), severity to family (r = 0.210), concern from NCT (r = 0.274), and trust (r = 0.116), all with a *p*-value < 0.001.

In general, exposure to risk communication significantly decreases the likelihood that a participant will indicate a behavioral intent to seek medical attention at the hospital, even when exposure to the rumors was intended to drive such behavior (Table 4). This is true for both types of news outlets: classic news websites and social media. Participants not exposed to risk communication on the classic media report indicate a mean intent score to seek medical attention of 3.48 (±1.10SD), 3.21 (±1.16SD), and 3.35 (±1.16SD) on the chemical, biological and radiological scenarios, respectively. This score was significantly higher than that reported by the participants in the parallel groups in which risk communication was offered: 2.88 ± 1.30 (t = −3.890, df = 240, *p* < 0.001), 2.89 ± 1.23 (t = −2.057, df = 237, *p* = 0.041), and 2.91 ± 1.23 (t = −2.865, df = 238, *p* = 0.005), respectively.

Intent to seek medical attention at the hospital in light of the NCT scenario presented was also found to be associated with the interest in additional information on NCT preparedness (*p* < 0.001), the overall effect of emergencies on the participant (r = 0.187, *p* < 0.001), and the total number of behavioral intents reported by a participant (r = 0.132, *p* < 0.001).

Participants were prompted to indicate which additional actions they would take in response to the NCT scenario in two cases: first, if the event took place as presented, and second, if it happened in their hometown (Table 5). For each participant, the total number of actions reported as intended to be performed (out of nine) was calculated. The mean score for the entire sample was 3.38 ± 2.06 when the scenario was as presented and 3.89 ± 2.31 if happening in the participant’s hometown. This difference is statistically significant according to the paired samples *t*-test (t = 12.748, df = 1801, *p* < 0.001). Within the chemical scenario group, the mean score rose from 3.18 ± 2.03 to 3.81 ± 2.25 (*p* < 0.001), within the biological scenario group from 3.29 ± 2.03 to 3.83 ± 2.26 (*p* < 0.001), and within the radiological scenario group from 3.67 ± 2.12 to 4.03 ± 2.41 (*p* < 0.001).

### 3.5. Predicting Behavioral Intent

An ordinal logistic regression analysis to investigate behavioral intent to seek medical attention at the hospital following an NCT scenario was conducted (Table 6). The predictor variables were tested a priori to verify there was no multicollinearity. The full model was a significant improvement in fit over the null model (χ^2^ = 173.903, df = 14, *p* < 0.001) and explains 15.6% of the total variance of the dependent variable. The results of the regression analysis indicate that the odds of participants exposed to risk communication to report an elevated intent of seeking medical attention were 0.470 (95% CI: 0.359, 0.615) times that of participants not exposed to risk communication, a statistically significant effect, Wald χ^2^ = 30.366, *p* < 0.001. An increase in threat intrusiveness was associated with an increase in the odds of an elevated intention of seeking medical attention, with an odds ratio of 1.295 (95% CI: 1.115, 1.505), Wald χ^2^ = 11.400, *p* = 0.001. Similarly, an increase in concern from NCT was associated with an increase in the odds of an elevated intention of seeking medical attention, with an odds ratio of 1.360 (95% CI: 1.195, 1.549), Wald χ^2^ = 21.630, *p* < 0.001. An increase in age was associated with an increase in the odds of an elevated intention of seeking medical attention, with an odds ratio of 1.009 (95% CI: 1.002, 1.017), Wald χ^2^ = 5.645, *p* = 0.018.

The analysis was repeated for each type of NCT scenario separately with similar results, albeit only for the chemical and biological scenarios. Of note, the results of the regression analyses indicate that the odds of participants exposed to risk communication to seek medical attention were lower than that of participants not exposed to risk communication and instructions in both scenario types: Chemical–Odds Ratio (OR) = 0.364 (95% CI: 0.226, 0.588), Wald χ^2^ = 17.077, *p* < 0.001; Biological–OR = 0.490 (95% CI: 0.306, 0.783), Wald χ^2^ = 8.884, *p* = 0.003. The model did not yield a similar effect of risk communication in the radiological scenario (OR = 0.631 (95% CI: 0.395, 1.011); Wald χ^2^ = 3.668, *p* = 0.055).

## 4. Discussion

The findings of this study refute the first hypothesis and support the second. The results do not indicate any substantial difference in threat perception components across different types of media outlets or in relation to exposure to rumors. Similar findings were reported by another study exploring Israeli public reaction to news sources in the context of earthquake preparedness [51]. According to the authors of that study, the type of media outlet had no significant influence on the perception of the earthquake threat or preparedness. The fact that the rumor (“fake news”) post did not generate an elevated threat perception could be attributed to the format in which it was administered, i.e., posted by an unknown person and with exaggerated language. Perhaps, participants “saw through the lie”.

Indeed, not all fake news is equal; however, some do take hold of the public imagination. While we may not believe an unknown person on social media, if it is repeated by a friend or family member we might be more inclined to believe it. For example, there are studies of radiological disasters that have shown people will wait to see what their neighbor does in an emergency to inform their own decisions and actions [14,52,53]. Nevertheless, we should also consider an alternative explanation proposed by the literature, which argues that users of social media are capable of utilizing mechanisms to question and criticize published content and refute rumors and ‘fake news’ [54].

In any event, there is reason to believe that when an NCT incident will happen, social media will be swamped with rumors and incorrect information [10,36], as is the case with the current COVID-19 outbreak [55]. Therefore emergency planners and risk communicators must anticipate misinformation in the wake of NCT and be prepared to refute it rapidly.

The results support the second hypothesis. Exposure to risk communication decreases significantly the intent to seek medical attention at the hospital in all study branches, as well as in the multivariate analysis. Risk communication was not only effective on its own, as demonstrated in the classic news report branches of the study, but also while coinciding with rumors calling people to seek medical attention, as demonstrated in the social media branches. The results here demonstrate the importance of risk communication in helping crisis managers to guide the public away from undesired behavior by the public.

The results of this section shed light and provide additional important evidence on several aspects. First, the results show no statistical difference between control and rumors (only) groups in their intent to seek medical attention (Table 4). This supports the assertion that participants might have been able to see through the rumor posted and were not affected by it in terms of increasing their intent to perform an undesired behavior. Conceivably, similar patterns of reaction may be expected in a real incident. Therefore, the aforementioned assertion that refuting rumors during a crisis is vital is further supported. In this sense, our study supports the findings reported by Simon et al. [36,54]. Second, the results demonstrate the uniqueness of the biological scenario. In this branch of the study, the only statistically significant difference observed following Bonferroni’s correction in decreasing intent to seek medical attention was between the risk communication group and the control. No significant difference was observed between the risk communication group and the rumored group, even though the former includes the rumor component in it. A possible explanation of these results may be in the contextual setting of this study during the Novel Corona Virus Disease (COVID-19) pandemic. Other studies have demonstrated multiple occasions of undesired behavior by the public during the COVID-19 outbreak, including panic buying [56,57] and disobeying quarantine-related health regulations [58]. This undesired behavior can be mitigated if public trust is maintained [40,59,60,61,62]. However, in Israel during the study period, there was a general environment of public dissatisfaction with governmental management of the COVID-19 crisis and distrust in public health decisions [63,64], which may have had residual effects on the measurements obtained from the biological scenario branch.

NCT threats have an extended component of uncertainty, which may elevate threat perception in comparison to other threats, such as disasters from natural sources or non-intentional man-made disasters [4,7]. Coupled with general distrust in the government’s actions, this can lead to an increase in undesired public behavior during the crisis [7,25,38,39,40,65]. To make things worse, the current study suggests that the ‘starting point’ of the proportion of Israelis who are likely to seek medical attention following an NCT incident is at about 45%. If eventuated, even a fraction of this figure could pose a serious challenge to hospitals dealing with a sudden influx of casualties from NCT. Nonetheless, the results also suggest that this intention can be attenuated with proper risk communication. Actually, the risk communication utilized in this study was quite simplistic and straightforward. It can be assumed that a more robust risk communication strategy may have an even greater effect to reduce undesired behavior.

It is important to note that this study assesses behavioral intent and we limit our discussion to this cognitive construct. There are several models designed to predict the translation of intentions into actual health behaviors, such as theory of planned behavior [66], theory of reasoned action [67], and the protection motivation theory [68]. However, these utilitarian models are based on assumptions that people use rational thinking when translating motivation into behavior, although we know this is not always the case with health behaviors [69]. Even dedicated behavioral models explaining specifically emergency preparedness behavior, such as those by Becker et al. [70] often resort to uncertainty as a general explanation to the multitude of occasions in which the models fail to predict behavior accurately. Therefore, assessing intent is a first step in understanding the behavioral phenomenon, but specific research is required to understand how said intent will eventually play out as behavior during real situations.

Planning for a proper risk communication strategy is also matched with the public desire to have more information. This result resonates with similar findings for other populations, e.g., the United Kingdom [14,42]. Given that a majority of the public is already aware of the NCT threat, this finding possibly suggests that commencing risk communication even during peacetime, i.e., before any NCT incident takes place, will not spark stressful reactions from the public [71]. Nevertheless, this should be done carefully under a well-crafted risk communication plan to ensure the desired public reaction to risk mitigation messages [8].

### Limitations

The primary outcome assessed in this study is behavioral intention under a hypothetical threat. Naturally, intentions do not always reflect actual behaviors; however, they provide a good estimate, especially when working assumptions have no real-world evidence to support them. As is the case with other cross-sectional studies, this study assesses measurements at a given point in time and under the circumstances existing in that period (in this case, the COVID-19 outbreak). One cannot rule out that assessment at other points in time will generate different results. It has also been noted already that the rumor post used in this study was designed to be from an unknown person to the participants. Different results may have been obtained had the rumor been presented by a family member or a close friend (a source of information known to be trusted by the recipient). Furthermore, while the tool utilized in this study was developed with input from experts in the field, it was not validated in a pilot study. This study also employed an online survey methodology. While a strength of this method is that it allowed for a large nationwide sample, the conclusions of this study cannot be generalized beyond people with computer access. Consequently, some constructs may be suboptimal in measuring the originally intended variables. Given the unique emergency circumstances existing in Israel, the generalization of the conclusion beyond the Israeli public should also be taken with caution. Another important limitation pertains to the study population. As the Israeli population is exposed to chronic terrorism, the generalization of the conclusions of this study to other populations exposed seldom to terrorism should be done with caution. Lastly, this study employed a univariate analysis that explored numerous associations between the studied variables. Although appropriate post-hoc corrections were made where necessary, this may also be seen as a limitation of this study.

## 5. Conclusions

This study demonstrates that behavioral intent to seek medical attention in hospitals following a non-conventional terror incident is attenuated by risk communication. Through proper guidance, instructions, and communication of the risks, crisis managers can decrease undesired behavior by the public and, consequently, gain more control over the situation. The findings of this study highlight the importance of establishing a risk communication strategy and devising plans to execute it in real-time, or even before, as a preemptive measure. In particular, this study suggests that overcoming fake news can be done with relatively simple risk communication, provided it is timely, clear, and provides people with instructions on how to secure their safety. Decision-makers should be encouraged to invest the necessary resources in creating a robust crisis communication infrastructure to mitigate the negative consequences of emergencies and amp up the support to crisis management. This effort should be aided by experts in risk communication, social psychology, and disaster management, to ensure the appropriate design of plans.

## Figures and Tables

**Figure 1 ijerph-19-00342-f001:**
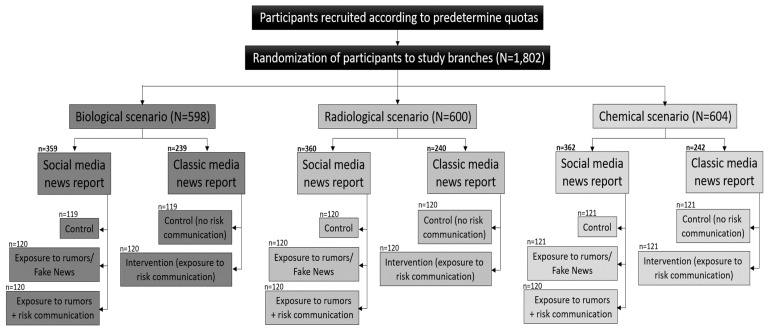
Flow chart of randomization of participants into study branches and groups.

**Table 1 ijerph-19-00342-t001:** Socio-demographic breakdown of study sample ^+^ (*N* = 1802).

Variable	*n* (%)	Variable	*n* (%)
**Gender**		**Religion**	
Women	927 (51.4%)	Jewish	1440 (79.9%)
Men	875 (48.6%)	Muslim	273 (15.1%)
**Age**		Christian	50 (2.8%)
Mean (±SD)	39.22 (±14.25)	Druze	39 (2.2%)
18–30	623 (34.6%)	**Affiliation to religion**	
31–45	599 (33.2%)	Secular	916 (50.8%)
46–55	295 (16.4%)	Traditional	492 (27.3%)
56–69	265 (14.7%)	Religious	241 (13.4%)
70+	20 (1.1%)	Ultra-orthodox	135 (7.5%)
**Place of residence (district)**		Missing	18 (1.0%)
Center	461 (25.6%)	**Education**	
North	338 (18.8%)	High school or less	300 (16.7%)
Tel-Aviv	293 (16.3%)	High school diploma	415 (23.0%)
Haifa	288 (16.0%)	Vocational studies	391 (21.7%)
South	208 (11.5%)	Bachelor’s	473 (26.2%)
Jerusalem	132 (7.3%)	Master’s or more	223 (12.4%)
Judea & Samaria	81 (4.5%)	**Income**	
**Birth Place**		Below average	812 (45.0%)
Israel	1591 (88.3%)	Average	375 (20.8%)
Elsewhere	211 (11.7%)	Above average	467 (25.9%)
		Missing	148 (8.2%)
**Family status**		**Occupation**	
Coupled w/children	973 (54.0%)	Employed (part or full time)	1098 (60.9%)
Coupled w/o children	301 (16.7%)	Self-employed	131 (7.3%)
Not coupled w/children	154 (8.5%)	Student	214 (11.9%)
Not coupled w/o children	374 (20.8%)	Unemployed/unpaid leave	151 (8.4%)
Mean No. family members (±SD)	3.88 (±1.88)	Military/national service	90 (5.0%)
Mean No. of children <18 (±SD)	1.29 (±1.91)	Retired	118 (6.5%)

^+^ No statistical differences were observed between the different scenario branches.

**Table 2 ijerph-19-00342-t002:** Mean scores of non-conventional terrorism threat perception components before and after intervention in all study branches (*N* = 1802).

Sample	Component	Before	After	t	*p*-Value
Overall sample (*N* = 1802)	Likelihood	2.42 ± 0.65	2.70 ± 0.96	14.235	<0.001
	Intrusiveness	2.73 ± 0.92	2.80 ± 0.92	3.666	<0.001
	Severity to society	3.10 ± 0.77	3.06 ± 0.77	−2.516	0.012
	Severity to family	2.94 ± 0.82	2.82 ± 0.80	−7.739	<0.001
	Concern	2.78 ± 1.18	2.91 ± 1.12	6.740	<0.001
	Trust	3.49 ± 0.98	3.40 ± 1.04	−6.231	<0.001
	Sense of preparedness	2.57 ± 0.91	2.61 ± 0.93	2.873	<0.001
Chemical scenario (*N* = 604)	Likelihood	2.42 ± 0.92	2.80 ± 0.93	11.298	<0.001
	Intrusiveness	2.77 ± 0.93	2.83 ± 0.91	1.938	0.053
	Severity to society	3.11 ± 0.75	3.02 ± 0.77	−3.576	<0.001
	Severity to family	2.96 ± 0.79	2.79 ± 0.82	6.15−	<0.001
	Concern	2.76 ± 1.16	2.92 ± 1.11	5.027	<0.001
	Trust	3.51 ± 0.92	3.41 ± 1.03	−3.774	<0.001
	Sense of preparedness	2.56 ± 0.89	2.60 ± 0.92	2.015	0.044
Biological scenario (*N* = 598)	Likelihood	2.40 ± 0.94	2.71 ± 0.93	9.131	<0.001
	Intrusiveness	2.68 ± 0.88	2.75 ± 0.89	2.127	0.034
	Severity to society	3.08 ± 0.76	3.03 ± 0.76	−1.890	0.059
	Severity to family	2.95 ± 0.83	2.82 ± 0.79	−4.688	<0.001
	Concern	2.76 ± 1.16	2.91 ± 1.10	4.393	<0.001
	Trust	3.43 ± 1.01	3.33 ± 1.04	−3.439	0.001
	Sense of preparedness	2.57 ± 0.93	2.58 ± 0.95	0.774	0.439
Radiological scenario (*N* = 600)	Likelihood	2.42 ± 0.98	2.58 ± 1.01	4.539	<0.001
	Intrusiveness	2.74 ± 0.95	2.82 ± 0.95	2.282	0.023
	Severity to society	3.11 ± 0.80	3.14 ± 0.79	1.198	0.231
	Severity to family	2.92 ± 0.84	2.85 ± 0.81	−2.605	0.009
	Concern	2.83 ± 1.23	2.91 ± 1.16	2.349	0.019
	Trust	3.54 ± 1.01	3.45 ± 1.06	−3.576	<0.001
	Sense of preparedness	2.59 ± 0.92	2.63 ± 0.94	2.156	0.031

**Table 3 ijerph-19-00342-t003:** Distribution of responses (*n*, %) to the question “if the news report you read took place in reality, what do you think were the chances that you would seek medical attention at the hospital?”.

Chance	Overall Sample (*N* = 1802)	Chemical Scenario (*N* = 604)	Biological Scenario (*N* = 598)	Radiological Scenario (*N* = 600)
Very little	196 (10.9%)	63 (10.4%)	66 (11.0%)	67 (11.2%)
Little	284 (15.8%)	99 (16.4%)	88 (14.7%)	97 (16.2%)
Somewhat	501 (27.8%)	157 (26.0%)	160 (26.8%)	184 (30.7%)
High	545 (30.2%)	189 (31.3%)	191 (31.9%)	165 (27.5%)
Very high	276 (15.3%)	96 (15.9%)	93 (15.6%)	87 (14.5%)

**Table 4 ijerph-19-00342-t004:** Differences in the mean score of the behavioral intent to seek medical attention at the hospital following social media exposure to non-conventional terrorism reports, rumors, and risk communication.

Study Group ^+^	Chemical Scenario			Biological Scenario			Radiological Scenario		
	Mean (±SD)	95% CI	F (*p*-Value)	Mean (±SD)	95% CI	F (*p*-Value)	Mean (±SD)	95% CI	F (*p*-Value)
I. Control	3.48 (±1.06)	3.29, 3.67	10.54 (<0.001)	3.59 (±1.15)	3.38, 3.80	3.98 (0.020)	3.36 (±1.15)	3.15, 3.57	7.65 (0.001)
II. Exposure to rumors	3.54 (±1.13)	3.33, 3.74		3.46 (±1.10)	3.26, 3.66		3.41 (±1.07)	3.22, 3.60	
III. Exposure to rumors + risk communication	2.92 (±1.27)	2.69, 3.15		3.17 (±1.29)	3.28, 3.53		2.88 (±1.27)	2.64, 3.11	
Bonferroni’s correction *	I vs. III: MD = 0.56 (±0.15SE), *p* = 0.001			I vs. III: MD = 0.42 (±0.15SE), *p* = 0.019			I vs. III: MD = 0.48 (±0.15SE), *p* = 0.004		
	II vs. III: MD = 0.62 (±0.15SE), *p* < 0.001			II vs. III: MD = 0.29 (±0.15SE), *p* = 0.172			II vs. III: MD = 0.62 (±0.15SE), *p* = 0.001		

^+^ Within the social media report study branch; * No differences observed between control (I) and rumor (II) groups across all scenarios (data not shown); MD = Mean Difference, SD = Standard Deviation, SE = Standard Error, CI = Confidence Interval.

**Table 5 ijerph-19-00342-t005:** Distribution of behavioral intents (% of top answers on Likert scale) in reaction to the non-conventional scenario presented and % of change (in brackets) if the scenario were to happen in the participant’s hometown.

Behavioral Intent	Overall Sample (*n* = 1802)	Chemical Scenario (*n* = 604)	Biological Scenario (*n* = 598)	Radiological Scenario (*n* = 600)	*p*-Value (χ^2^)
Stay tune for more info on the media	70.3 (−4.5)	70.2 (−3.6)	70.2 (−3.8)	70.5 (−6.0)	0.992 (0.016)
Contact family members	49.4 (+4.9)	48.3 (+3.9)	50.0 (+6.2)	49.8 (+4.7)	0.819 (0.400)
Stock supplies of food and water	48.6 (−4.3)	46.0 (−1.0)	46.0 (−3.7)	53.8 (−8.1)	**0.007** (9.814)
Contact the emergency call center	42.2 (+8.1)	38.6 (+9.4)	45.0 (+9.0)	43.0 (+6.0)	0.070 (5.309)
Shelter in place and close windows and AC	38.4 (+8.1)	36.1 (+12.1)	33.1 (+10.0)	46.0 (+2.2)	**<0.001** (23.084)
Avoid sending the kids to school for some time	35.6 (+9.5)	30.8 (+12.2)	36.8 (+8.9)	39.2 (+7.5)	**0.008** (9.784)
Avoid going to work for some time	28.1 (+10.9)	23.5 (+14.1)	29.3 (+8.0)	31.7 (+10.5)	**0.005** (10.468)
Use the gas mask	12.9 (+4.6)	12.3 (+4.1)	9.9 (+5.2)	16.5 (+4.7)	**0.002** (12.064)
Evacuate somewhere far away	12.5 (+13.7)	12.3 (+11.9)	9.0 (+14.1)	16.3 (+15.0)	**0.001** (14.633)

Bolded *p*-values indicate statistical significance below a *p*-value of 0.05 (two-tailed).

**Table 6 ijerph-19-00342-t006:** Results of ordinal logistic regression analysis to predict behavioral intention to seek medical attention at the hospital following a non-conventional terrorism scenario (*N* = 1081).

Variable	B (SE)	Wald χ^2^	*p*-Value	OR	95% Wald Confidence Interval for OR	
					Lower	Upper
Gender (0-female, 1-male)	0.062 (0.116)	0.286	0.593	1.064	0.848	1.336
Age (cont.)	0.009 (0.004)	5.645	0.018	1.009	1.002	1.017
Effect of emergencies *	−0.006 (0.755)	0.007	0.932	0.994	0.857	1.152
Perception of Likelihood *^+^	0.018 (0.073)	0.060	0.806	1.018	0.882	1.175
Threat intrusiveness *^+^	0.259 (0.077)	11.400	0.001	1.295	1.115	1.505
Severity to society ^#+^	0.026 (0.098)	0.068	0.794	1.026	0.846	1.244
Severity to family ^#+^	0.169 (0.109)	2.411	0.121	1.185	0.957	1.467
Concern from NCT *^+^	0.308 (0.066)	21.603	<0.001	1.360	1.195	1.549
Trust *^+^	0.175 (0.060)	8.627	0.003	1.192	1.060	1.340
Sense of preparedness ^+^	0.102 (0.067)	2.312	0.128	1.108	0.971	1.264
Exposure to rumors (0-no, 1-yes)	−0.063 (0.135)	0.219	0.640	0.939	0.721	1.222
Exposure to risk communication (0-no, 1-yes)	−0.755 (0.137)	30.366	<0.001	0.470	0.359	0.615
Interest in more information (0-no, 1-yes)	−0.156 (0.119)	1.734	0.188	0.855	0.678	1.079
Total no. of behavioral intents (cont.)	0.047 (0.285)	2.686	0.101	1.048	0.991	1.108

* ordinal 5-point Likert scale; ^#^ ordinal 4-point Likert scale; ^+^ Post-intervention measurements used.

## Data Availability

Data is available upon reasonable request.

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
