# Peer review of "The Effect of Risk Communication on Public Behavior to Non-Conventional Terrorism—Randomized Control Trial"

_ijerph, 2021, doi:10.3390/ijerph19010342_

Round 1

Reviewer 1 Report

The work presented is very good and completed by an acceptable number of individuals.

The scenarios raised seem correct to me and within plausible situations.

Weren't you raised at some point to compare with the conventional attack?

In the socio-demographic study, the authors observe religion as a variable. It is not clear to me that this variable determines in an emergency situation whether or not they go to a health center.

One point that I see more negative about the study is the population. It is a population accustomed to warfare. The response to conventional and unconventional attacks may be conditioned by this fact. Would this same study, carried out in geographic regions with fewer war conflicts, give the same answer?

In the conclusion, they should talk a little about how to work with fake news. Let it be clearer since today with the ease of issuing comments that reach society so quickly, how to cut that route.

There are three references that I have not found in the text

6 Mohtadi, H., Murshid, A.P. Risk analysis of Chemical, Biological, or Radionuclear Threats: Implications for Food Security. Risk 583 Analysis: An International Journal 2009, 29(9), 1317-1335. 584

8 Ruggiero, A., Vos, M. Communication Challenges in CBRN Terrorism Crises: Expert Perceptions. Journal of Contingencies and 587 Crisis Management 201523(3), 138-148.

33 Beaton, R., Stergachis, A., Oberle, M., Bridges, E., Nemuth, M., Thomas, T. The Sarin gas attacks on the Tokyo subway-10 years 628 later/Lessons learned. Traumatology 200511(2), 103-119. 

Author Response

We thank the reviewer for his/her important comments. Please see the modifications made to the manuscript in the attached file.

Reviewer 2 Report

This is a well-presented study, however, there are major issues, as follows:

-For this study, the novelty is completely not clear.

-The research hypothesis and methods are not described well. As well as the main contribution is completely confused.  

-Additionally, the number of samples is less than expected. So, additional samples are needed to build strong conclusion about this problem

Author Response

(The authors gave the same response as above.)

Round 2

Reviewer 2 Report

The authors response to all comments.